# Analysis of Commuting Distances of Low-Income Workers in Memphis Metropolitan Area, TN

**Anzhelika Antipova**

Earth Sciences Department, University of Memphis, Memphis, TN 38152, USA; antipova@memphis.edu

**Abstract:** The paper tests whether low-income workers suffer a greater commuting cost burden compared with a typical commuter within the context of decreasing economic opportunity. The paper adds to the spatial mismatch research by studying the metropolitan area in the U.S. South, which experienced "some of the largest decreases" in job proximity in 2012. Memphis, Tennessee, saw the disproportionately steep declines in the average employment opportunities within a typical commute distance experienced by low-income and minority residents. The paper first delineates low-income neighborhoods across the study area, then identifies commuting patterns within the three-state study area including the greater Memphis, and lastly, it compares average commute lengths by a typical and a low-income commuter, as well as the shares of resident workers with a long commute by earning category. The paper offers insight into the ways in which the changes in spatial location of employment and population within the metropolitan area may impact commuting distance for disadvantaged low-income travelers. We show low-income workers commute statistically significantly shorter distances to their places of work compared with a typical commuter. Our other results find that disadvantaged workers in Shelby County, TN, are disproportionately concentrated in lower-wage industries, such as hospitality and retail service industries, compared to overall workers. Finally, a significantly greater proportion of disadvantaged workers travel long distances of over 50 miles compared with higher-earning workers, indicating the disparity in commuting patterns between a typical resident and a low-income worker.

**Keywords:** commuting; spatial mismatch; low-income worker; low-income tracts; Memphis metropolitan area; Shelby County

## 1. Introduction

Considerable attention has been given to regional economic growth plans in the Southern U.S., such as what types of jobs and development policies can drive long-range growth [1]. One strategy to economic growth includes eliminating poverty and increasing employment, and improving social mobility via increased transportation access due to its substantial effect on the latter, which is even more significant than factors such as crime, education, and family structure [2]. In the U.S. context of suburbanized jobs with dispersed employment subcenters, to organize public transit to dispersed workplaces is a challenge [3]. In sprawling urban areas such as Memphis, where due to historical patterns of decentralization of population and wealth, the poor is a substantial share of the City of Memphis' population; the link between poverty and transportation is augmented due to a disconnection of low-income residents from entry-level jobs by distance and by inadequate public transit transportation systems providing no access to suburban jobs or amenities [4]. Recent research shows that even with the explosive growth of alternative mobility service providers, such as transportation network companies (TNCs), claimed to improve urban mobility and transportation equity, there tend to be fewer Uber pickups in low-income neighborhoods, at least in the case of New York City [5].

Southern urban areas in the USA witnessed changes in both population and their economies, which resulted in specific spatial patterns and economic structures resulting in an increasingly polycentric area with multiple newly emerged downtowns which successfully compete with the central business district (CBD) [6]. Growing distance between people and jobs in metropolitan America negatively affects accessibility and challenges regional economic development within the Southern U.S. Increasing commuting impacts economic and social outcomes, including local financial viability due to the fiscal tax base derived from commercial and industrial taxes, the quality of public services, residential proximity to retail, and employment outcomes for residents, especially among low-income and minority workers [7]. The latter is related to the duration of joblessness for black, female, and older workers.

Parallel to changes in the urban spatial structure of metropolitan areas, there are substantial changes to the workforce and workplaces of metropolitan areas, including increasing job polarization of the new economy characterized by growth of high- and low-wage jobs and decline in many middle-wage jobs, and continuing gender inequality and job insecurity.

The spatial location of jobs within metropolitan areas matters as residents in different parts of the area may be affected differently by shifts in the job distribution. Better job accessibility, measured as a proximity to jobs, increases the likelihood of working, especially among the disadvantaged residents whose job prospects are often limited by jobs near their residential neighborhoods due to lower transportation access and other resources, and thus may face spatial barriers to jobs when appropriate jobs relocate to suburban areas [8].

An overall employment increase alone does not necessarily guarantee an increase in the number of jobs near the typical resident. Between 2000 and 2012, in most of the nation's largest metropolitan areas, job opportunities have decreased within the typical commute distance, more so for suburban residents than for central city commuters, despite overall metropolitan employment growth. This happened because suburban jobs became more spread out, so that average job density in the suburbs declined even though the total number of jobs increased. Besides inequality due to geographic barriers to employment (that is, job disparity between city and suburbs), non-spatial factors including inequality in commuters' socioeconomic status (such as race, education, income level, industry sector) matters in job accessibility [9]. Poor and minority residents experienced a steeper decline in proximity to jobs (where job proximity is measured as the number of jobs proximate to each census tract, or how many jobs fall within the median commute distance between the origin and destination tract within each metropolitan area) compared with non-poor and white residents [7].

As jobs shifted from the urban core outward, a residential shift continued, including suburbanization of minorities and the poor. The disadvantaged groups, due to the legacy of restrictive zoning and housing policies, have traditionally had lower suburbanization rates compared to white and non-poor residents. However, for the first time in 2010, the disadvantaged population residing in the suburbs across the country's largest metropolitan areas have finally exceeded those living in the urban core.

As poor and minority residents suburbanized in the 2000s, they experienced a greater decline in proximity to jobs compared to non-poor and white residents. Jobs tend to decline in areas of poverty and minority resident concentrations. That is, regardless of their location in the central city or suburbs, for the most disadvantaged commuters, the number of jobs within a typical commute distance declined at a faster pace during the 2000s than it did for non-disadvantaged residents of other communities. Especially marked declines in job proximity have been experienced by the residents of high-poverty census tracts (where the poverty rate is at least 20 percent) and majority-minority neighborhoods (i.e., where non-Hispanic Whites account for less than 50 percent of residents) [7].

Prior research explored the link between home and work locations, starting with the spatial mismatch hypothesis introduced by [10]. Within this strand of work, spatial mismatch literature investigated the impact of job suburbanization and wealth concentration away from the central city on employment outcomes for disadvantaged residents trapped in the central city. Recent spatial mismatch studies examine residential neighborhood characteristics [11], observing growing numbers of

low-income and minority households in inner-ring suburbs with older housing and aging infrastructure poorly served by public transportation and thus facing lengthy commutes, effectively rendering jobs in the distant suburbs out of reach, as well as in outer-ring suburbs, similarly with no access to well-paid jobs unless one possesses a private automobile [12–15], and residence-based job accessibility [16]. Studies commonly use commuting distance as an outcome of job accessibility [9]. Commuting flow data are thus needed to compare commuting distance between typical commuters and those from the disadvantaged neighborhoods, referred to interchangeably in this research as low-income tracts.

The current stream of literature examines commuting inequalities [12] and the results can be used for the development of transportation and housing policies. Socio-spatial polarization emerges in post-Fordist cities, where a spatial and temporal variation in high- and low-income groups' out-of-home activities exist with different activity spaces among high- and low-income people [17]. Low-income individuals' out-of-home activities, including workplaces, tend to group close to their residential spaces and high-income clusters are more likely to be away from their home locations. As mentioned above, in the U.S. context, job decentralization over several decades resulted in some jobs clustered in subcenters (better served by public transit than dispersed workplaces) and other jobs uniformly dispersed in a low-density fashion, with the debate continuing regarding the degree of a difference between commutes to decentralized and centralized workplaces due to workers' potential adjustment of their job and housing locations and stabilization of commute distance or time [18]. Regarding the type of workplace, low-income workers tend to work in centralized and dispersed workplaces, versus high-income workers who mostly work in employment clusters, partly explaining shorter-distance commutes and more public transit use by lower-income workers [18]. In a New York City-based study examining temporal trends in commuting and wage differentials by race and gender, many minority workers, particularly minority women, experienced a decline in access to jobs [12], while in another report, many low-wage, hourly-paid employees had long commutes to their low-paid jobs in New York City [19].

Another recent study explored how the average commuting distance is related to income at the individual level, focusing on Denmark, the UK, and the USA, observing good agreement between the model and empirical data for these different countries [20]. Using the 2009 National Household Travel Survey (NHTS) dataset, each worker was associated with an income category (personal income was substituted by the household income divided by the household size) and the one-way distance to workplace. As income increases, there is a slow increase of the average commuting distance, while for large distances there is a slowly decaying tail. Another observation of commuting patterns among the U.S. lower-income workers is that despite shorter average commute distances, this group of the labor force has a longer commute time [21,22]; among all gender and race groups, Black male and female workers suffer from persistent spatial mismatch reflected in the longest commuting times [12]. Overall, all full-time workers had a consistent increase in the average weekly commuting time since 1990, which increased by almost one-half hour [19,23] linked employment clusters with commutes of different income groups. Additionally, recent studies look at the relationship between travel characteristics and satisfaction with travel; for example, [24] examine the psychological impact of commuting focusing on the low-income working population—lower-income respondents are less satisfied with their commuting. Commuting satisfaction is influenced by commuting characteristics and the attitudes towards travel, however, various factors contribute to satisfaction among high- and low-income groups differently—travel attitudes towards specific travel modes are related in a significant way with commuting satisfaction in the latter group.

The paper's main objective is to understand whether the evidence of the spatial mismatch hypothesis can be found for disadvantaged travelers (here understood as low-income) within the context of the overall net job losses, and the decreasing job opportunities within the typical commute distance for all commuters. The disadvantaged are understood in this research as minorities and the poor in the Memphis metropolitan area. The spatial mismatch is tested using commuting length as a yardstick. The paper uses a three-step approach. First, low-income residential areas need to be

identified using a consistent rationale and criteria; second, commuters from low-income residential tracts have been matched to low-wage workplace locations (also referred to as low-paying jobs) to determine a realistic commuting distance by a low-income worker. The third step is comparing commuting lengths between a typical commuter and a low-income commuter.

The paper is organized as follows. The first section is an introduction and the subsequent section presents the study area. Next, we report data used in this paper and the methods applied in the third section. The fourth section explains how low-income neighborhoods and tracts with low-paying workplaces were identified, and how low-earning commuters have been matched to low-wage jobs. The same section also presents jobs by distance analysis. Finally, findings can be found in the Results of the analysis of commuting distances section. We close by providing conclusions and a discussion of the future work in the sixth section.

## 2. Study Area

Memphis metropolitan area is located in the U.S. South. According to the Bureau of Economic Analysis' [25] definition, the state of Tennessee is included in the region of the U.S. South. Overall, the South is becoming increasingly diverse in terms of industry mix, economic output, and performance [26]. In Tennessee, regarding industrial makeup, healthcare is the leading industry for the high-growth firms [1], while the Memphis metropolitan area has diversified the economic base, where service-providing industries comprise 89% and goods-producing industries account for 10.7%, all depending on the distribution and transportation that the region offers [6]. Its urban form was shaped by aggressive annexation and suburban sprawl, resulting in the total area of over 324 square miles (845 square km), twice the area of Denver (154 square miles), but less than half Denver's population density (1997 vs. 3882 per square mile, [27]). The thinly spread population accounts for the public transit system's inability to connect residents to jobs; Memphis was ranked 69th among the top 100 metropolitan areas on all metrics of transit accessibility, which consisted of coverage, service frequency (minutes), and job access, reflecting the poor intra-metropolitan accessibility of jobs via transit within the area [28]. The shifting locational advantage from the urban core outward resulted in changes in the urban spatial structure of its metropolitan area. Within major cities, such as New York, jobs are concentrated in the CBD and distant suburbs [12]. In the study area, the greatest employment concentration can be found 11 miles away from the Memphis' CBD (within this distance jobs consistently exceed labor), after which jobs tend to decrease in Shelby County, however, with other job spikes located at 23, 35, and 40 mi. away from the CBD, indicating suburban job locations where the largest employing industries consist of construction, retail, administrative support, and professional and scientific sectors [6].

Population suburbanization and concomitant wealth decentralization led to an increasing inequality between parts of the region. Income inequality intensifies social differentiation, making income an important index for social stratification [17]. Florida [29] ranked the Memphis metropolitan area the second highest regarding the income segregation level (that is, the tendency of various income groups to live separately from other groups) across large metropolitan areas. Income segregation increased with income inequality [30], which in turn, is exacerbated by a continuing Black–White residential segregation. Despite declining segregation of residential locations among Blacks and their growing movement to suburban areas, their housing remains highly segregated from that of Whites. These spatial residential and job patterns together with inefficient transit, a high degree of job sprawl, and a high dependence on automobiles have resulted in a spatial mismatch between where low-income residents live and where jobs are available. Recent studies observe emerging social diversity in people's out-of-home activities in addition to that in the residential sector [17]. Accordingly, better understanding of mobility patterns, which is a central problem for policymakers and planners in the interdisciplinary science of cities [20], provided motivation for the research, and we focus here on a low-income segment of the labor force, specifically, on the issue of how land use arrangement may impact low-income workers in terms of their commuting distances.

The Memphis metropolitan area saw a pattern in employment change similar to that described above for the U.S. largest metropolitan areas. Memphis experienced "some of the largest decreases" in job proximity [7] p. 5.

The U.S. metropolitan areas which had the greatest increases and decreases in job proximity, 2000–2012, were selected for the comparison in Figure 1. Among the 96 largest metro areas, the 6th largest decrease in total metropolitan jobs occurred in Memphis, TN-MS-AR. From 2000 to 2012, total jobs fell by 7% across the metro area, as shown in Figure 1. With employment shifting toward suburbs during the 2000s, there was a decrease in job proximity, or the number of jobs near both a typical city and a suburban resident. The Brookings Institution measured proximity as the average number of jobs within the typical metro area commute distance, weighted by census tract population [7]. Using this indicator of employment opportunity, the average jobs near a typical resident decreased from 180,154 in 2000 to 149,539 in 2012, or a change of 17% across the study area.

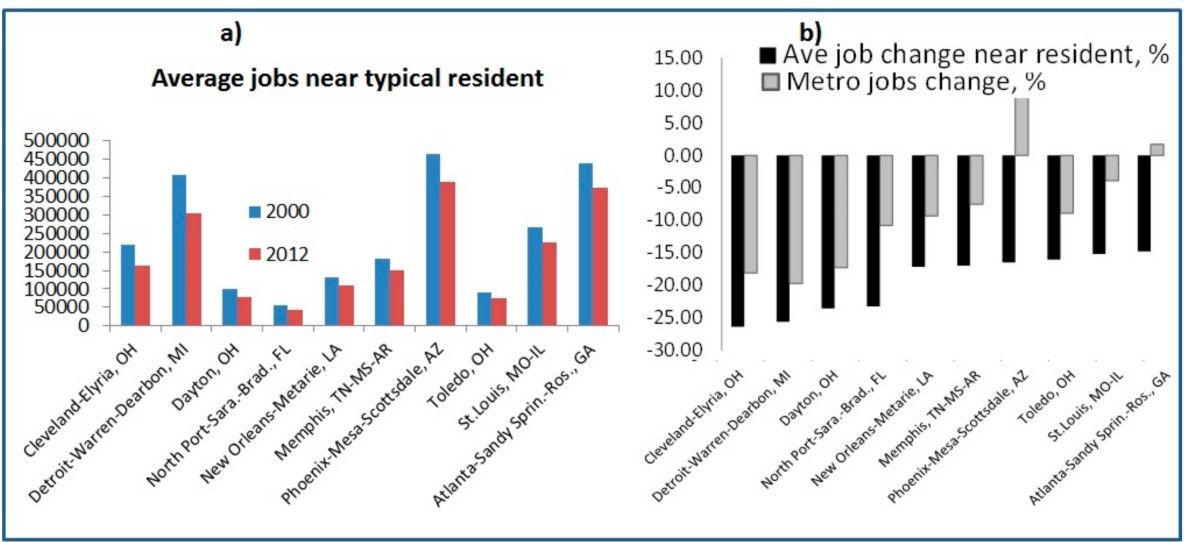

**Figure 1.** The U.S. metro areas with the greatest increases and decreases in job proximity, 2000–2012: (a) Average jobs near a typical resident; (b) Average percent job change near residents and an overall metropolitan job change. Source: [7].

## 3. Data

This section describes data used in the study, including (1) commuting flow data, (2) the U.S. Census tract geography (boundaries) data, (3) local labor and employment data, and (4) 2012–2016 American Community Survey (ACS) 5-year estimates.

Several sources provide data for this type of research. First, the tract-to-tract flow data available in the most recent 2012–2016 Census Transportation Planning Products (CTPP) provide total worker counts commuting from residence tracts to workplace tracts. The data are based on the 2012–2016 American Community Survey (ACS) 5-year estimates of commuting. The information is organized by residence, workplace, and by the commute from home to work. A survey is conducted annually by the U.S. Census Bureau. Using the survey-based data, changes in the socioeconomic, housing, and demographic characteristics of communities can be examined across the United States [31]. The ACS travel-related questions concentrate only on commuting, no information is available on other travel purposes, such as leisure or other nonwork travel. Data is available on workers 16 years and older and who were employed during the ACS reference week. Respondents are asked how they get to work and may choose from among several transportation modes. Data of community flows by travel mode is available at a county level for the entire country. However, to understand the nature of commuting at the intra-county level, a finer scale of commuting flows is needed, making the ACS-based data less suitable.

The Census Transportation Planning Products (CTPP) allow for analyzing commuting flows (home to work) at various geographies including a tract level. The dataset is organized by where workers live (Part 1), where they work (Part 2), and by the flow between them (Part 3). Demographic characteristics are included in residence- and workplace-based tables and flow tables for home to work, such as age, race, sex, earnings, income, employment status, industry, occupation, household size, vehicle availability, means of transportation to work, and a host of others. Based on the 2012–2016 5-year ACS, the CTPP are useful in studies trying to understand where people are commuting to and from, and how they get there. The information is organized by residence, workplace, and by the commute from home to work. The file used in this study contains all tract pairs which present flows from the CTPP 2012–2016 (https://ctpp.transportation.org/2012-2016-5-year-ctpp/). Residence and workplace state, county, and census tract Federal Information Processing Standard (FIPS) codes which uniquely identify geographic areas in the United States are also provided.

The CTPP-based data at a tract level provide journey-to-work trips for all commuters. Crosstabs of workers with selected demographic and travel characteristics are also available in the dataset. For this study, we used the CTPP-based data at a tract level of journey-to-work trips for all commutes without distinguishing a mode of transportation using one measure in flow data which is the total count of workers and its associated margins of error. However, since commuters mostly rely on automobiles (according to the American Community Survey 2012–2016, over 92% of commutes within the study area was made by private motorized vehicles, as shown in Figure 2), the dataset is considered suitable for the studies investigating the spatial distribution of employment and labor and commuting patterns to answer questions such as where workers commute to work and what is the commuting trip length distribution. The 2012–2016 5-year American Community Survey (ACS) data are designed to help transportation analysts and planners understand where people are commuting to and from, and how they get there. Using this data, one can obtain an origin–destination (OD) matrix for the study area under investigation by extracting all records for those census tracts representing residential places ("from-tracts"). To determine commuting characteristics (the average trip length) for low-income commuters, one needs to identify low-income tracts. A detailed description of the method used to identify low-income neighborhoods across the study area is given below, but we provide a brief outline here. Because we are focusing on low-income tracts, we first use all residential tracts within the Memphis metropolitan area to determine a typical commute distance, while to do the same for low-income commuters, we limit the entire sample to residents of tracts meeting our criteria of low income by requiring that over 33% (that is, 1 standard deviation above the average level) of tract residents have earnings at a certain level.

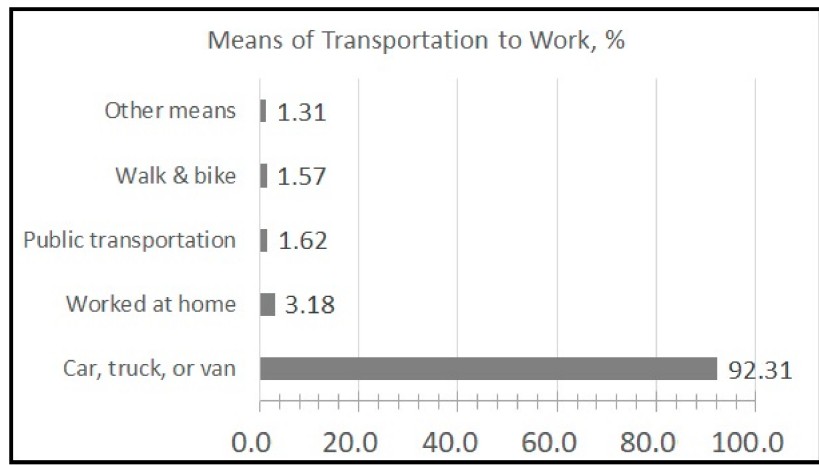

**Figure 2.** Commuting in Shelby County, TN, by travel mode for 2013–2017. Source: American Community Survey (ACS) 2013–2017. Workers 16 and older. ACS 5-year estimates, table B08301.

The second type of data used by the spatial mismatch research is produced by the Longitudinal Employer-Household Dynamics (LEHD) program at the U.S. Census Bureau (interchangeably referred to here as the *OnTheMap* flow database, or LED) providing data on residence and workplace for many categories of work travelers including their earning level utilized in this research.

1. To understand commuting patterns across the study area, we used tract-to-tract flow data. As described above, this file includes census tract-to-census tract flows from the Census Transportation Planning Products (CTPP) 2012–2016. Previous studies noted the importance of the size of the job market used for the analysis of job accessibility [32–35]. When job market size is artificially truncated and is equated with the residential locations, residential locations at the edge of the study area have low job accessibility, termed *frontier effects* [36]. To address this problem, larger job markets are recommended [9]. Accordingly, to avoid frontier effects and to understand the extent of spatial mismatch (if any) for low-income residents in Shelby County, Tennessee, this research uses surrounding tracts in Tennessee, Mississippi, and Arkansas as the potential job market for Shelby County residents in the determination of commuting distance. We extracted all commuting flows from all the tracts in Shelby County, Tennessee (County FIPS = 157, State FIPS = 47) to all tracts in Tennessee (State FIPS = 47), Arkansas (State FIPS = 28), and Mississippi (State FIPS = 05). Using this data, we obtained an origin–destination (OD) matrix of commuting flows within the metropolitan area of Memphis, TN-MS-AR. There were 10,977 flows in total between tracts of residence in Memphis and workplace tracts in Tennessee, Mississippi, and Arkansas in total.

2. To understand if there is any difference in commuting measured by distance in miles between an average commuter and a disadvantaged commuter measured by earnings, commute distances have been compared between respective commuter groups in Shelby County, TN. We used both the 2016 LED and the 2012–2016 CTPP data. The LED data are organized by where workers live ("Home") or where workers are employed ("Work") by various segments of the labor market, such as worker age, earnings, race, gender, and education. We used earnings data, that is, workers who earn $1250 or less per month. Using the LED-based worker earnings data on the home end, we have identified low-income residential tracts in Shelby County (TN) (N = 42). To extract commuting flows, we used the CTPP 2012–2016 Part 3 flow data at the level of Census tracts, which provides the identification numbers of the origins (home tracts) and destination (work tracts) of journey-to-work travel. The tract-to-tract flow data available in the 2012–2016 Census Transportation Planning Products (CTPP) provide total worker counts commuting from residence tracts to workplace tracts. Commuting flows have been extracted from all residential tracts in Shelby County (for N tracts = 221, there are 10,977 flows for 2012–2016) and also just from low-income home tracts (for N tracts = 42, there are 1034 flows for the same period) to all work tracts, for which there were commuting flows within the study area covering states of Tennessee, Arkansas, and Mississippi with and without matching low-income commuters to low-wage workplace locations (see subsequent sections below for low income identification rationale and methodology).

3. For the analysis of commute distance, all tracts which generated commute trips have been used, including those tracts which produced work trips in the same home tract. Commute trips within the same census tract have been included since, as per land use and transportation policy goals, locating jobs near homes is a desired outcome to reduce greenhouse gases from vehicle miles traveled (VMT) which measures the total amount of driving. Excluding these trips may introduce bias in the calculation of average commuting distance. To compare mean commuting distances between an average commuter and a low-income traveler (without matching to low-income employment), a statistical two-sample *t*-test assuming unequal variances has been conducted (the results can be found in the Results section).

4. Since commuting flow data are available for the level of tracts, accordingly, this study used the geographic files providing boundaries of the study area at the tract level for 2016. Cartographic

boundary files are available in shapefile format from the U.S. Census. Cartographic boundary shapefiles of Census tracts for 2016 for the entire state of Tennessee, Mississippi, and Arkansas were obtained from the TIGER products offered by the U.S. Census Bureau (https://www.census.gov/geo/maps-data/data/cbf/cbf_tracts.html). Substantial research continues analyzing employment subcenters due to their impact upon labor markets and commuting patterns [18]. The location of residential areas and the activity locations, such as work, are important "anchors" of daily activity spaces [17]. Accordingly, we extracted 221 residential tracts located within Shelby County, TN.

5.　To identify low-income neighborhoods in Shelby County, we used data from *OnTheMap* (interchangeably referred to here by the name of the program that produced the data, or Local Employment Dynamics, or LED data), a web-based mapping and reporting application located at https://onthemap.ces.census.gov/. We explain the rationale and a step-by-step description of low-income residential tract identification below.

　　While socioeconomic factors, different transportation mobility, and residential location constraints (such as social and institutional barriers, and housing unaffordability) substantially impact commuting patterns of various population groups, workplace locations are the fourth factor that contribute to commuting inequalities with workers in various income categories getting sorted into different workplace locations ([18]. The *OnTheMap* data can be used for the analysis of the spatial organization of the local labor force and employment. The application was developed by the U.S. Census Bureau and its Local Employment Dynamics (LED) partner states, showing workplace (where workers are employed or "Work") and residential distribution (where workers live or "Home") at census block level in 50 partner states/territories, with consecutive years of data from 2002 to 2017. Besides worker and employment locations, it also provides data on age, earnings, industrial sector, race, ethnicity, educational attainment, and sex for both Home and Work locations. Data on the "Home" end was downloaded for all workers who resided in Shelby County in 2016. Since this data was in the format of point locations (representing census blocks), the points were spatially joined to census tracts within the geographic information system (GIS) environment so that each tract was given the summed numeric values of block-based points that fell inside it.

6.　To verify the selected low-income tracts, we checked the selection against areas where households had income in the past 12 months below poverty level using the 2013–2017 American Community Survey (ACS) 5-year estimates (https://www.census.gov/geo/maps-data/data/tiger-data.html). However, since the ACS-based poverty data comes at a block group scale, the data was used only for cross-validation purposes.

## 4. Identification of Low-Income Residential Tracts

　　This section describes in detail how the above data have been used in the study, including (1) an identification of low-income residential tracts, and (2) an analysis of commuting measured by distance in miles for all commuters and low-income workers with and without matching of low-income commuters to low-wage jobs.

　　Prior research examined the spatial and temporal clustering of the high- and low-income groups' activities [17]. We used LED data to identify the location of low-income residential workers within the study area. An area can be analyzed based on where workers live ("Home") or where workers are employed ("Work"). Residential location of workers was implemented by performing home area profile analysis. The data provides an output in the form of points which represent census blocks, which can be further analyzed using geographic information system (GIS) software. In total, 12,347 points representing 12,347 census blocks where workers resided in 2016, have been added to ArcMap and aggregated to 221 census tracts by spatially joining points to tracts, so that the summed numeric values of block-based points that fell inside tracts, were given to tracts' attributes. Table 1 reports that there were 410,174 resident workers who lived in Shelby County tracts in 2016 based on the LED data used to identify low-income residential tracts (330,955 resident workers in total have been identified

who commuted for work from Shelby County tracts to all tracts in TN, MS, and AR using the 5-year estimated data from the CTPP averaged for the period of 2012–2016. We used this data for analyzing commuting length distribution).

**Table 1.** Resident workers by earnings, for all jobs, Shelby County, TN, 2016. Source: LED data.

| Earning level | Count | Percent |
|---|---|---|
| <=$1,250 | 108.437 | 26.4 |
| $1,251-$33,333 | 149,014 | 36.3 |
| <$3,333 | 152,723 | 37.2 |

Low income can be measured by applying federal poverty thresholds for different-sized families. For example, a family is considered a low-income family if the family's income falls below 200 percent of the federal poverty level (that is, for a family of four, including two children under 18 years, whose poverty threshold level in 2010 was established at $22,314, the family's income has to be below $44,628 a year). To identify low-wage commuters, McLafferty and Preston [12] used the Public Use Microdata Sample (PUMS) data on hourly wages self-reported by individuals, and wages were combined into quartiles based on these hourly wages. We used monthly earnings developed by the LED to define a low-income worker. Earnings are grouped into three categories: (1) $1250 per month or less; (2) $1251 to $3333 per month; (3) more than $3333 per month.

A worker who earns $1250 per month or less is defined as a low-income worker. This value is close to the amount of the poverty-level wage earned by a full-time worker monthly developed by the Economic Policy Institute (EPI) using government data sources [37]. Using the EPI-based poverty-level wage threshold in 2010 of $10.63 per hour (wages are in nominal dollars) and assuming 7 h worked per day, and 5 days worked per week, the employment yields monthly earnings ranging from $1488 ($10.63 × 5 × 7 × 4) to $1674 ($10.63 × 5 × 7 × 4.5) calculated for a 4-week and 4.5-week paying period, respectively. Workers who work 35 h or more a week are considered full-time workers [38]. However, setting the wage threshold for low-wage workers at 150 percent of the federal minimum wage, which was $7.25 per hour in 2010 [39], for full-time employment may yield slightly higher values of $1522 and $1712, for 4-week and 4.5-week paying periods, respectively.

Census tracts were designated as low-income tracts when the percent of workers of the lowest earning category was one standard deviation above the average level, that is, those tracts where at least 33.59% (26.92% + 6.67%) of the tract's workers earned $1250 or less per month. Forty-two tracts out of 221 in total met this designation criterion as a residential low-income neighborhood. The map in Figure 3a presents the location of low-income tracts identified using the methodology described. The identified residential pattern of low-income workers in Shelby County is in agreement with prior research findings regarding residential location of low-wage workers in low-income families with children who are more likely residing in central cities, suggesting only limited access to better-paying jobs in the growing suburban areas among low-income workers [38].

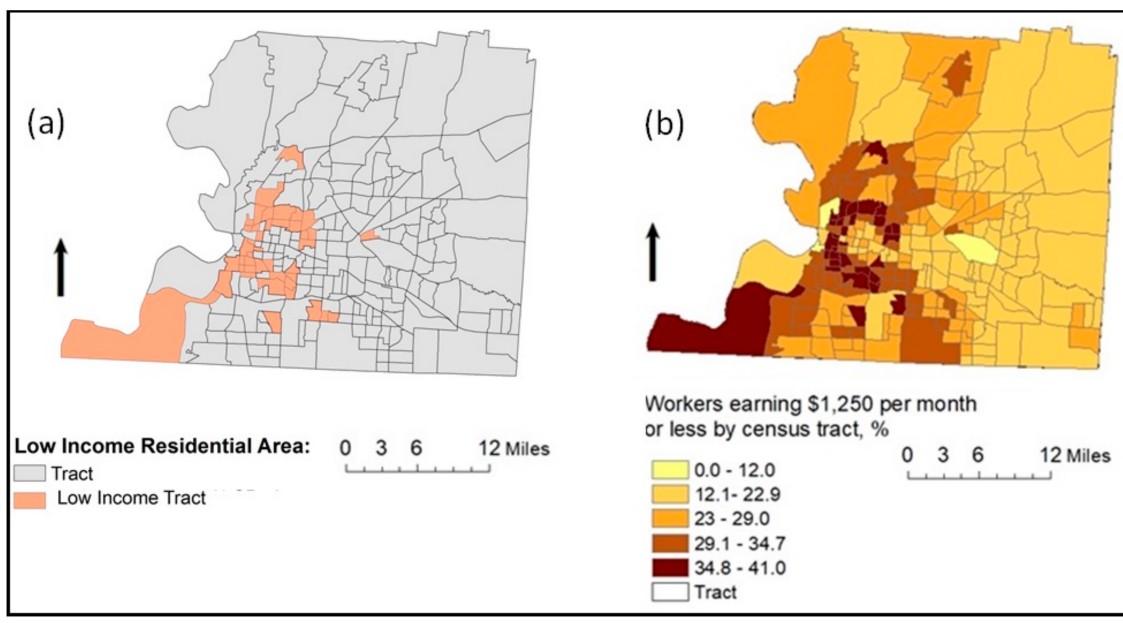

**Figure 3.** (**a**) Low-income residential areas (42 tracts used in the study); (**b**) distribution of workers earning $1250 per month or less by census tract in Shelby County, TN, 2016.

Figure 3b visualizes the distribution of workers earning $1250 per month or less by census tract in Shelby County, TN, 2016. In this study, to be consistent with commuting flow data at the tract level, we used 42 low-income tracts identified using the LED worker earnings data, as shown in Figure 3a. Low-income tracts were used as the "from" tracts (that is, origin low-income tracts) when analyzing commuting distances for low-income commuters.

## 4.1. Identification of Low-Paying Work Tracts and Matching Low-Earning Commuters to Low-Paying Work

We also matched workplace locations for commuters from low-income residential tracts in Shelby County. The study used an approach similar to that used by Morton et al. [40], who simulated likely place of work for worker categories whose residential locations and workplace industries were known. By using the reported journey to work tract-to-tract mean travel flows by industry, the number of employees working for each business in every industry in each block group has been inferred by summing them up and calculating the probability that a randomly selected resident living in a certain home tract and working in a certain industry works in a block group. Given that commute distribution is known (from the 2012–2016 CTPP tract-to-tract flow data), we also need to know how many jobs a certain population segment could reach that match their skill set within a given commute shed. In order to match workplace locations, we used the following approach. We combined the Census Transportation Planning Products (CTPP) journey-to-work tract-to-tract mean travel flows (CTPP 2012–2016) with employment data provided by the LED. More specifically, we first identified locations of low-paying jobs across the states of Tennessee, Arkansas, and Mississippi using the LED-based job data by worker earnings on the work end (that is, only the low-paying job market segment, or employment paying $1250 per month or less was used) and exported these obtained shapefiles of work areas to the GIS aggregating low-paying jobs at the tract level. This was done by using each business point's earning level data, and then summing over the number of employees working for each business in every census tract within a particular earning category (here, with wages of $1250 or less per month). Second, to quantify observed patterns, we used a spatial pattern analysis, specifically, a statistical method of Hot Spot Analysis (Getis-Ord Gi*) with 90, 95, and 99 percent confidence levels. The Hot Spot Analysis evaluates the spatial arrangement of features and uses a statistical measure (that is, the Getis-Ord Gi* statistic with Z-scores, or standard deviations, and *p*-values, or statistical probabilities) to find areas with statistically significant concentrations of a phenomenon of interest,

here low-wage employment. Generally, hot spots represent areas where observed spatial patterns are not due to random processes, but they are rather locations where underlying spatial processes are at work [41]. Thus, we identified concentrations of low-paying jobs across all tracts in the study area. The hot spots (shown in red color for the Memphis metropolitan area) in Figure 4 denote low-wage hot spots found in 854 tracts out of 2847 total tracts in the study area (not shown in the map), as well as the location of low-income residential tracts in Shelby County. The identified low-wage clusters contained 535,038 low-income jobs in total, capturing 41.7% of a total 1,283,634 low-paying jobs in the 3-state area of Tennessee, Arkansas, and Mississippi (out of 2,211,546 paying jobs in these 854 tracts in total).

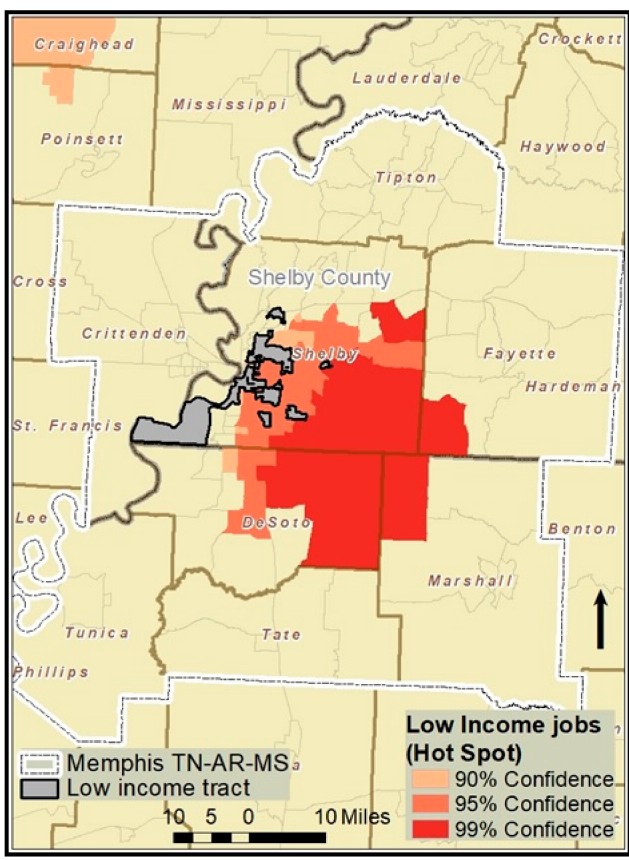

**Figure 4.** Low-income hot spots in the Memphis metropolitan area within the study area in 2016.

Table 2 reports the job market structure by earnings within the study area separately by state and the USA, 2016. The average percent of low-paying jobs is 30% per work tract and a standard deviation of 11.5% in the low-wage hot spot areas, and 28.3% per tract on average across all 2847 tracts and a standard deviation of 10.9% within the study area (not shown in Table 2). To take into account the underlying employment, in addition to being a hot spot cluster, more restrictive conditions have been imposed to better match low-income commuters to most likely employment. With this in mind, to ensure that at least one third of commuters are employed in a low-wage sector, we also identified tracts where the share of low-wage jobs was at least 33%. Using the same rationale, we identified work tracts where low-paying jobs were above one standard deviation (40%), at least half of all jobs (50%), and 60 and 66 percent of all jobs (however, there were either no or a very few flows from low-income tracts to the last two categories of employment, so we show no information for those work tracts). We used several combinations of the above in order to match workplace locations for commuters from low-income residential tracts in Shelby County: (a) tracts with at least 33% low-wage jobs, (b) hot spots of low-wage jobs, (c) tracts with at least 40% low-wage jobs, (d) tracts with at least 50% low-wage jobs, (e) tracts which are hot spots and which have at least 33% low-wage jobs, (f) tracts which are hot

spots and which have at least 40% low-wage jobs, (g) tracts which are hot spots and which have at least 50% low-wage jobs. Commute flows have been extracted from low-income tracts to the work tracts meeting the above specifications. Commute distances have been compared between low-income commuters travelling from residential low-income tracts to work tracts with and without matching to appropriate employment and all commuters in Shelby County for nine various scenarios using a single factor ANOVA (the results of the analysis can be found in the Results section).

**Table 2.** All jobs by earnings within the study area and the USA, 2016. Source: LED data.

| Jobs by Earnings, per Month/Count | TN | % | AR | % | MS | % | TN, AR, MS: Total | TN, AR, MS: % | USA: Total | USA: % |
|---|---|---|---|---|---|---|---|---|---|---|
| $1250 or less | 691,017 | 24.1 | 287,914 | 24.3 | 304,703 | 27.4 | 1,283,634 | 24.83 | 32,709,154 | 23.5 |
| $1251 to $3333 | 1,073,707 | 37.4 | 484,638 | 40.9 | 464,247 | 41.7 | 2,022,592 | 39.13 | 46,942,884 | 33.8 |
| More than $3333 | 1,105,787 | 38.5 | 413,629 | 34.9 | 343,587 | 30.9 | 1,863,003 | 36.04 | 59,407,326 | 42.7 |

*4.2. Jobs by Distance Analysis*

The 2012–2016 CTPP data require matching to appropriate employment to realistically understand commuting distance for a particular labor market segment, such as low-income commuters. However, the LED jobs by distance data can be extracted for a desired group of commuters, thus, requiring no matching, providing absolute and relative job counts broken out by four commute distance categories (less than 10 miles, 10 to 24 miles, 25 to 50 miles, and greater than 50 miles). It can answer a question of what distances do local workers commute? The analysis outputs data of the distance totals between residence and employment locations for workers living in the study area and employed elsewhere. The data are useful to examine the movement of workers from their homeplaces to their workplaces. Figure 5 presents job count data by commuting distance from home areas both in low-income tracts and in the entire Shelby County to work tracts by labor market segment (that is, separately for all commuters and low-income commuters, i.e., workers earning $1250 per month or less) using the LED jobs by distance data, 2016, providing both relative (above the bars on the chart) and absolute (inside the bars) values. It can be seen that consistently many more low-income workers travel long distances (greater than 50 miles) compared with an average commuter (11.7% vs. 9.5%).

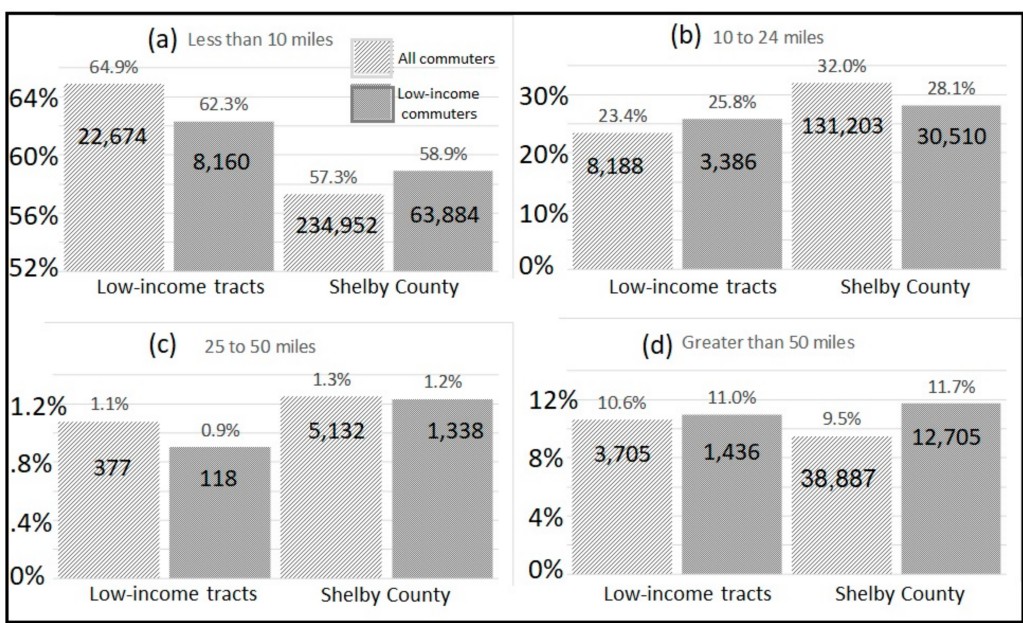

**Figure 5.** Commuter shares by distance for average and low-income commuters, 2016: (a) jobs within less than 10 miles of commuting; (b) jobs within 10 to 24 miles of commuting; (c) jobs within 25 to 50 miles of commuting; and (d) jobs at greater than 50 miles of commuting. Source: Local Employment Dynamics (LED) jobs by distance data.

## 5. Results of the Analysis of Commuting Distances

Figure 6 illustrates commuting flow characteristics, including commuter count, distance, and direction of journey-to-work travel by the Shelby County population. Specifically, commuting flows produced by commuters from all residential tracts and low-income tracts in Shelby County, to work tracts in Tennessee, Arkansas, and Mississippi are visualized. Figure 6a illustrates commuting flows for all commuters from the residences in Shelby County to all tracts in Tennessee, Mississippi, and Arkansas, which were also used to help identify where low-income populations travel for work, as shown in Figure 6b. We compared travel distance distribution for all workers and that for low-income workers.

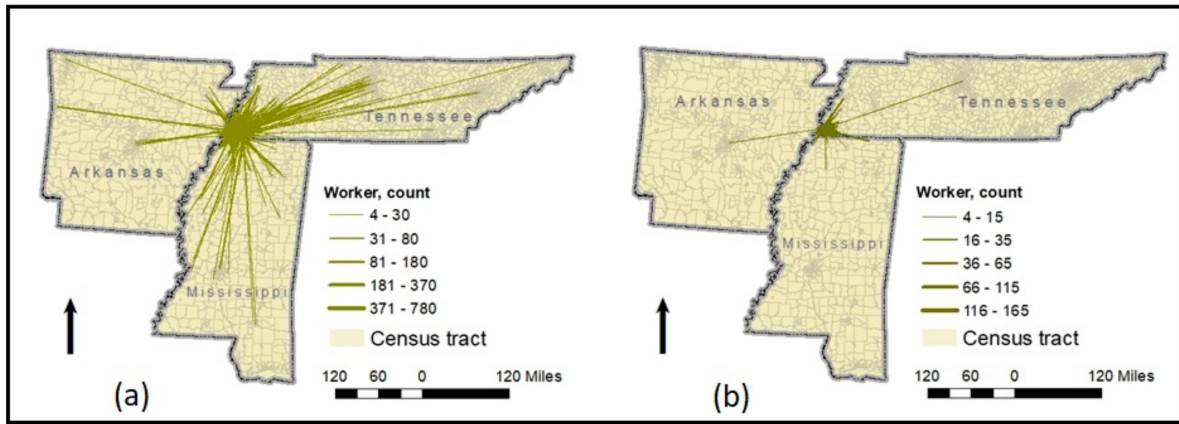

**Figure 6.** Commuting flows by commuters from (**a**) all residential tracts and (**b**) low-income tracts in Shelby County, 2016, to all tracts in TN, MS, AR. Source: Census Transportation Planning Products (CTPP) 2012–2016.

We were interested in the number of commuters to understand commuting structure by distance. Based on the CTPP 2012–2016 tract-to-tract flow data, Figure 7 shows distribution of commute distance in miles by workers residing across all tracts in Shelby County (a) and those in low-income tracts (b), without matching to potential low-paying jobs. It is essential to provide jobs close to places of residence, especially for disadvantaged workers, so we compared the percent of commuters from both commuter groups (from all tracts and from low-income areas) who work in the same tract of home residence (reflected on the chart by the "0" distance). Importantly, many more average commuters work within the same tract where they live than low-income workers (8.3% vs. 5.9%), as shown in Figure 7. In agreement with a recent study of the effects of distance on commuting flow and a determination of a commuting threshold where commuting distance starts negatively affecting commuting flows [9], we observe the distribution of commuting distances conforming not to the exponential pattern, but being more right-skewed.

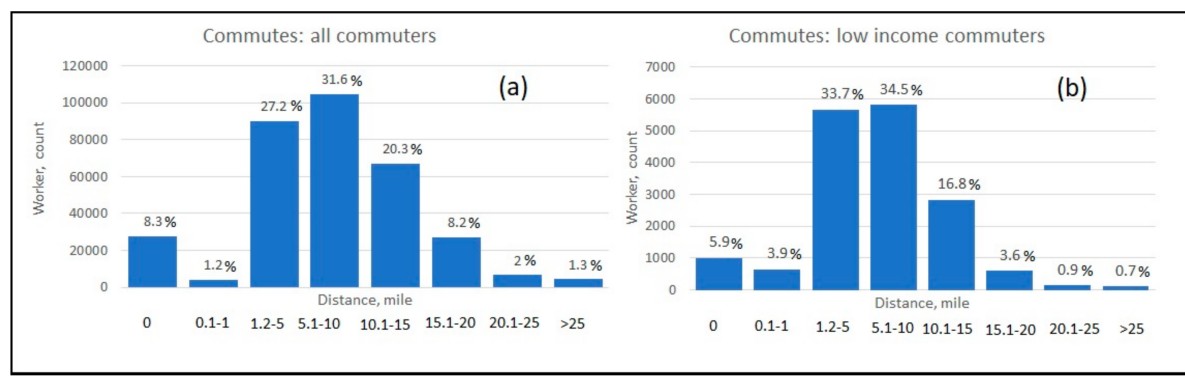

**Figure 7.** Commute trip distribution by length to tracts in TN, AR, MS for: (**a**) all commuters; and (**b**) low-income commuters. Source: CTPP 2012–2016 data.

Table 3 provides descriptive statistics of all commuting distances in miles using tract-to-tract CTPP flow data. The average distance is 10.01 miles. An identified average commute distance comes close to the typical commute in miles of 8.9 miles determined by the Brookings Institution for the Memphis metro area, TN [7]. Besides reporting a commuting length for a typical commuter, Table 3 shows an average distance of 7.8 miles by a low-income commuter. We implemented a two-sample *t*-test assuming unequal variance for the commuting distance study, as shown in Table 4.

**Table 3.** Commuting length for a typical and low-income commuter, miles. Source: 2012–2016 CTPP data.

| N, Worker Count | Mean Length, Miles | Std Dev | Std Error | Min | Max |
|---|---|---|---|---|---|
| All tracts: 330,955 | 10.01 | 14.83 | 0.02 | 0 | 411.2 |
| Low Income tracts: 16,663 | 7.8 | 9.6 | 0.0857 | 0 | 202.5 |

**Table 4.** The results of the *t*-test: two-sample assuming unequal variances. Source: 2012–2016 CTPP data.

| | Distance in Miles, All Commuters | Distance in Miles for Low-Income Commuters, No Matching |
|---|---|---|
| Mean | 10.01 | 7.79 |
| Variance | 219.9 | 92.09 |
| Observations (flows) (N jobs) | 10,977 (330,955) | 1034 (16,663) |
| Hypothesized Mean Difference | 0 | |
| df | 1543 | |
| t Stat | 6.71 | |
| P(T ≤ t) one-tail | <0.0001 | |
| t Critical one-tail | 1.65 | |
| P(T ≤ t) two-tail | <0.0001 | |
| t Critical two-tail | 1.96 | |

Alpha = 0.05.

Table 5 presents the results of the single factor ANOVA of commuting distances by commuters from low income and all residential tracts with and without matching to low-income jobs. The results are reported for nine various scenarios. Means are significantly different—a typical commute distance of 10.01 miles agrees with the previous finding by the Brookings Institution for the Memphis metro area, TN, which is 8.9 miles. We show that low-income workers commute statistically significantly lower distances to workplaces compared with a typical commuter (reflected in low p-values, as shown in Table 4).

**Table 5.** Single-factor ANOVA of commuting distances by commuters from low income and all residential tracts with and without matching to low-income jobs. Source: 2012-2016 CTPP data.

| From-Tracts | Matched to Low-Income Jobs | To-Tracts | Flows, Count | Commuters, Total | Mean Distance, Mile | Variance |
|---|---|---|---|---|---|---|
| Low income residential tracts in Shelby County (N = 42) | yes | With at least 33% Low-Income jobs | 257 | 3818 | 8.85 | 182.84 |
| | | Hot Spots of Low-Income jobs | 879 | 14,325 | 7.38 | 83.17 |
| | | With at least 40% Low-Income jobs | 126 | 1725 | 7.62 | 22.08 |
| | | With at least 50% Low-Income jobs | 30 | 417 | 8.33 | 17.19 |
| | | Hot Spots and with at least 33% Low-Income jobs | 902 | 14,567 | 7.52 | 85.10 |
| | | Hot Spots and with at least 40% Low-Income jobs | 884 | 14,368 | 7.41 | 83.11 |
| | | Hot Spots and with at least 50% Low-Income jobs | 30 | 417 | 8.33 | 17.19 |
| | no | All work tracts in study area | 1034 | 16,663 | 7.80 | 92.09 |
| From all tracts in Shelby County (N = 221) | no | All work tracts in study area | 10,977 | 330,955 | 10.01 | 219.91 |

| ANOVA | | | | | | |
|---|---|---|---|---|---|---|
| Source of Variation | *SS* | *df* | *MS* | *F* | *p-value* | *F crit* |
| Between Groups | 17,564.97 | 8 | 2195.6 | 11.92 | <0.0001 | 1.94 |
| Within Groups | 2,782,482 | 15,110 | 184.14 | | | |
| Total | 2,800,047 | 15,118 | | | | |

Figure 7 uses the CTPP 2012–2016 data and shows proportion of jobs located within specified distance bands for the typical worker and workers residing in low-income neighborhoods. For example, less than 6% of workers from low-income areas commuted within the same residential areas compared to the typical worker (over 8%). Within the average commuting distance of 7.8 miles, about three quarters of jobs are available for workers residing in low-income neighborhoods within the average commuting distance of 7.8 miles. The average traveler has access to 68% of jobs within the average commuting distance of 10 miles. The LED data allows a researcher to conduct a distance analysis for a desired labor market segment without having to match it to workplace locations and directly extract data from home areas under investigation. This option outputs data as a categorical variable in four ranges (e.g., jobs within less than 10 miles, 10–24 miles, 25–50 miles, greater than 50 miles). Gabe et al. [42] used midpoints of the ranges when a variable under consideration was categorical. Similarly, for the purposes of estimating the number of miles travelled during the commute, we used midpoints of the ranges for the number of miles travelled. Table 6 reports the distribution and the average commuting distance for low-income residents to low-wage jobs compared to the other two income/wage categories.

**Table 6.** The distribution and the average commuting distance for low-income residents to low-wage jobs, compared to the other two income/wage categories in Shelby County, TN. Source: 2016 LED data.

| Home Area | Low-Income Tracts | Shelby County Tracts |
|---|---|---|
| Earning level | Mean Distance, miles (N= jobs available by earning type) | |
| Low-earning | 14.42 (13,100; 37.5%) | 15.22 (108,437; 26.4%) |
| Medium-earning | 13.3 (15,951; 45.6%) | 14.01 (149,014; 36.6%) |
| High-earning | 14.92 (5893; 16.7%) | 14.35 (152,723; 37.2%) |
| Total | 13.99 (34,944) | 14.46 (410,174) |

Table 7 uses the LED data and shows job shares by distance and by wage type. While the average distance is shorter for low-wage workers, the proportion who commute long distances is higher. For example, 11% of low-earning workers from low-income residential tracts commute longer distances (11.7% of workers from the same earning category residing across Shelby County tracts), higher than the average commuter from low-income residential tracts (10.6%), and the average commuter residing across Shelby County tracts at 9.5%.

**Table 7.** Proportion of jobs within specified distance bands for workers of low-income tracts by wage type in Shelby County, TN. Source: 2016 LED data.

| Home Area: Low-Income Tracts | | | | | | | | |
|---|---|---|---|---|---|---|---|---|
| Distance | All Workers | | Low-Income Workers | | Medium-Earning Workers | | High-Earning Workers | |
| Total All Jobs | 34,944 | 100.00% | 13,100 | 100.00% | 15,951 | 100.00% | 5893 | 100.00% |
| Less than 10 miles | 22,674 | 64.90% | 8160 | 62.30% | 10,682 | 67.00% | 3832 | 65.00% |
| 10 to 24 miles | 8188 | 23.40% | 3386 | 25.80% | 3571 | 22.40% | 1231 | 20.90% |
| 25 to 50 miles | 377 | 1.10% | 118 | 0.90% | 172 | 1.10% | 87 | 1.50% |
| Greater than 50 miles | 3705 | 10.60% | 1436 | 11.00% | 1526 | 9.60% | 743 | 12.60% |
| Home Area: Shelby County Tracts | | | | | | | | |
| Distance | All Workers | | Low-Income Workers | | Medium-Earning Workers | | High-Earning Workers | |
| Total All Jobs | 410,174 | 100.00% | 108,437 | 100.00% | 149,014 | 100.00% | 152,723 | 100.00% |
| Less than 10 miles | 234,952 | 57.30% | 63,884 | 58.90% | 89,391 | 60.00% | 81,677 | 53.50% |
| 10 to 24 miles | 131,203 | 32.00% | 30,510 | 28.10% | 44,047 | 29.60% | 56,646 | 37.10% |
| 25 to 50 miles | 5132 | 1.30% | 1338 | 1.20% | 1854 | 1.20% | 1940 | 1.30% |
| Greater than 50 miles | 38,887 | 9.50% | 12,705 | 11.70% | 13,722 | 9.20% | 12,460 | 8.20% |

We also implemented a job distribution by the North American Industry Classification System (NAICS) Industry Sector, which classifies business establishments by type of economic activity, separately for all paying jobs and low-wage paying jobs for resident workers of Shelby County, TN, 2016. To match low-income workers to jobs by wage level, we extracted the low-wage paying segment of the labor market. For consistency, jobs where workers earned $1250 per month or less were designated low-wage jobs. There were 108,437 jobs meeting this condition in 2016. Figure 8 presents the distribution of (a) all paying jobs by NAICS Industry Sector, and (b) low-wage paying jobs by NAICS Industry Sector, Shelby County, TN, 2016. We found a higher concentration of low-wage workers in Shelby County, TN, in lower-wage industries such as hospitality and retail service industries, compared to higher-earning workers, as shown in Figure 8. Overall, 21.3% of all jobs were in accommodation and food services, 20% of the total were in waste management and remediation, 16.7% of all jobs were in retail trade, 16.75% were in health care and social assistance, while 6.3% were in transportation and warehousing, as shown in Figure 8.

Figure 9 presents employment by distance in miles travelled by high-, medium-, and low-earning workers using the LED data. We tested the hypothesis of the equality of three proportions obtained from independent samples using the Pearson chi-square test. Although earlier we did not find statistical evidence of a greater *commuting length* by low-income residents compared with other earning categories, a chi-square test of proportions revealed that a significantly *higher share of low-income workers* (11.7%) have long commutes compared with either medium earning (9.2%), or the highest earning group of commuters (8.2%), ($X^2$ = 955.5714, *p* < 0.0001), as shown in Table 8. Figure 9 shows a greater proportion of low-income residents who have to commute long distances (defined as a distance over 50 miles), which is an indicator of spatial mismatch for this category of workers. However, higher wages do not seem to alleviate the social inequity caused by long commutes [43].

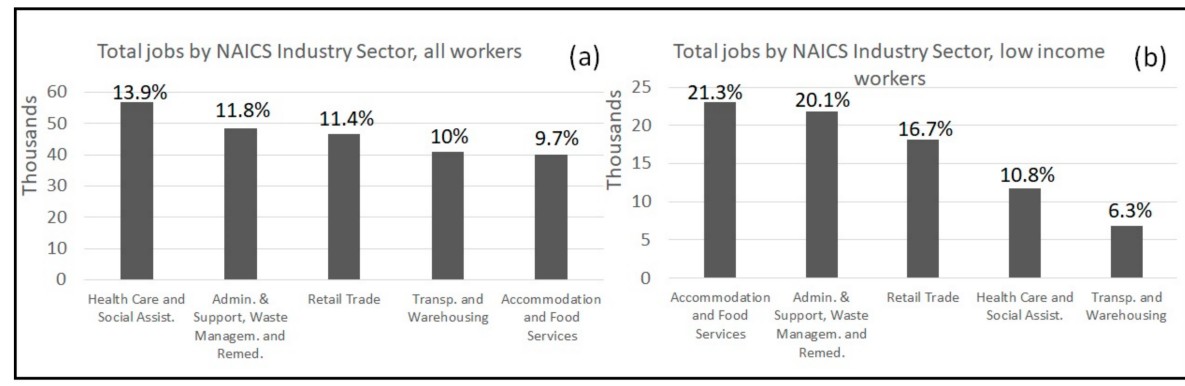

**Figure 8.** Distribution of jobs by NAICS Industry Sector in Shelby County, TN, 2016: (**a**) total jobs of all workers; and (**b**) total jobs of low-income workers.

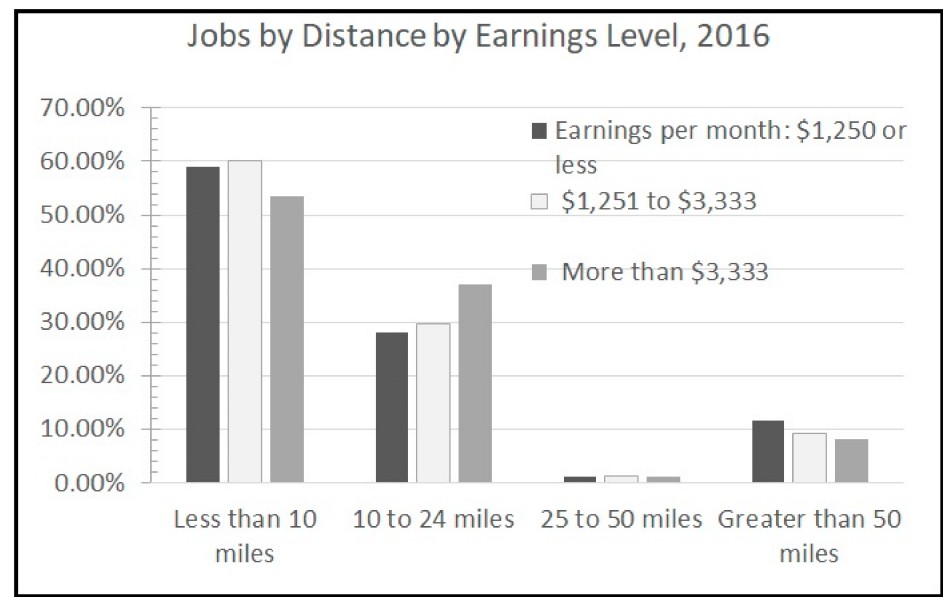

**Figure 9.** Employment by distance in miles travelled by high-, medium-, and low-earning workers.

**Table 8.** Results of a Pearson chi-square test of proportions of long commutes by worker income level in Shelby County, 2016.

| Worker Income Level | Jobs within Less 50 Miles | Jobs within More 50 Miles | Total |
|---|---|---|---|
| Low Income | 95,732 | 12,705 (11.7%) | 108,437 |
| Medium Income | 135,292 | 13,722 (9.2%) | 149,014 |
| High Income | 140,263 | 12,460 (8.2%) | 152,723 |
| Total | 371,287 | 38,887 | 410,174 |
| Statistics | Value | Probability | |
| Chi-Square | 955.5714 | <0.0001 | |

## 6. Discussion and Conclusions

Residents and communities within metropolitan areas have the diverse experiences of economic opportunity. Within regions, residential distribution patterns vary for various races and socioeconomic status resulting in great variations in job proximity across the groups. Within major metropolitan areas, in the period 2000–2012, residents experienced a decline in employment opportunities within the typical commute distance by seven percent. Especially marked declines in job proximity occurred for

commuters from high-poverty (where at least 20% of the tract's residents are poor) and majority-minority communities [7].

The paper looks for the evidence of the spatial mismatch hypothesis among disadvantaged commuters. Specifically, in the paper we investigated whether spatial mismatch measured by commuting distance in miles occurs among low-income commuters in Memphis, TN-MS-AR. In this research, we used the *OnTheMap* (also, referred to as the LED)-based lowest earning category as a measure of workers who are low income. Scholars in various fields utilize the LED-based data including the spatial mismatch hypothesis, encompassing a wide range of research questions, including commuting distance sensitivity and job accessibility, with a focus on low-wage workers [9], the effect of the spatial mismatch on joblessness among displaced workers [44], commuting among older adult workers [45], and regional research including trends in migration [46]. The LED-based data provide no trip characteristics (such as a mode, route, travel time, or departure time). A known problem with the data is inaccuracies in employment site locations that might occur when multiple worksite employers report all employees located at a primary employer address, and not a secondary worksite that is closer to their home location, resulting in a potential misallocation of workers to an employer's primary worksite [47]. A researcher needs to possess solid local knowledge to assign workers to worksites for multi-site employers. Another issue is missing employment categories, including federal, military, and railroad workers (1%–20%). For a description of other sources of publicly available, free-for-download datasets based on linked employer–employee records, which are useful for researchers in labor economics and regional science, a reader is referred to Manduca [48].

Based on the CTPP 2012–2016 data we analyzed commuting length distribution, a typical commuter travels on average over 10 miles to get to a place of work, while a commuter from low-income neighborhoods travels 7 miles. The typical commute identified for the study area is one mile longer compared with findings by the Brookings Institution for the Memphis metro area, TN, which is 8.9 miles (based on older data of the CTPP 2006–2010).

Compared with higher-income families, low-income families have shorter commutes on average [49]. In agreement with prior research, our findings indicate that low-income workers travel the shortest distances closer to the residential neighborhoods. However, distance sensitivity, or how far workers want to commute, exists among various groups of poor residents caused by socioeconomic characteristics, industry sectors, and personal choices affecting their employment outcomes, including job search range and job accessibility [9]. The authors found that distance might begin to affect residents' job choices only after a certain threshold, such as after 3 km, and subgroups of workers could have different commuting distance sensitivities [9].

However, the average statistics may mask significant variations within the population under investigation. When we looked at long commutes (that is, over 50 miles), our results found that a significantly greater proportion of disadvantaged workers travel long distances compared with higher-earning workers. It means a substantial share of disadvantaged workers for various reasons have employment outside of their residential locations, indicating the disparity in commuting patterns between a typical resident and a low-income worker.

Compared to the average worker, low-wage workers have different employment patterns. Low-income families earn insufficient money to pay for basic living expenses [49]. Low-income workers are more likely to be concentrated in industries with lower average hourly wages. Employment of low-wage workers in lower-wage industries, such as leisure and hospitality and other service industries, is higher than among workers in general [38]. This low-wage employment typically offers limited job advancement opportunities, few (if any) benefits, and less flexibility, creating challenges for parents trying to balance work and family responsibilities [49]. Service-related jobs often require working long hours and on nights or weekends, producing substantial challenges for childcare. Other sectors like health services, construction, and transportation and utilities, tend to pay better wages.

By adjusting affordable housing locations, or adding extra jobs near housing, a balanced jobs–housing relationship can be achieved to effectively shorten low-income workers' commuting

times [50,51]. Although greater job accessibility alone does not translate into better employment outcomes, moving jobs closer to low-income workers may alleviate spatial mismatch and decrease the share of workers with long commutes. Being closer to more jobs matters more for certain groups of workers. Poor and minority residents have transportation constraints, limited access to work supports including childcare, and a smaller job seeker's search area, posing greater barriers to connecting to economic opportunity compared to higher-income commuters [8,52].

To help decrease these barriers, a city may adopt policies decreasing commuting cost burden successfully implemented elsewhere, including the income-based policy in King County, in Seattle, where residents living below twice the federal poverty level have discounted public transit fares.

Since no information on commuting workers' commute choices which is available in the 2012-2016 CTPP flow data was used in this study, we made an assumption they all commute by car [44]. In the census tracts where our sample resides, over 92% of workers use private motorized vehicles, while other modes account for either 1% or less than 2%, somewhat justifying our assumption. Future studies may overcome this study's limitation and provide an in-depth understanding of low-income commuting by various travel modes.

Finally, similarly to other studies [7], straight-line distance was used in this study for the analysis of the commuting distance. However, freeways are disproportionately located in low-income neighborhoods and communities of color, potentially impacting distance calculations. In order to address this issue, future studies might use a real-world road network to measure shortest-path distances for more robust distance calculations.

**Funding:** This research received no external funding.

**Conflicts of Interest:** The author declares no conflict of interest.

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
