# Peer review of "Analysis of Commuting Distances of Low-Income Workers in Memphis Metropolitan Area, TN"

_sustainability, doi:10.3390/su12031209_

Round 1
Reviewer 1 Report
This paper examines the commenting distances of low-income workers in Memphis, TN. Authors first delineated the low-income neighborhoods, then identified the commuting patterns and compared average commute length by low-income commuters. Although the research is interesting and provides useful insights I found repetition in the Introduction, data and method sections. Paper is also not well organized. My specific comments and suggestion are below:
The intro is good but it also provides information on Data and limitations. I suggest moving Line 76 to 112 to the Data section. Provide only a summary here and merge details in the data section. Similarly, most of the information on Line 113-131 provides limitations. It should be in the discussion section. Analysis and results are mainly based on OnTheMap Are there any other studies where the same data were used? What are the pros and cons of using this data? Also, OnTheMap was mentioned so many times in the text that readers start to get irritated. Explain it once; use OnTheMap once or twice and then refer it as data. Un-necessary details are provided when it comes to basic GIS analysis such as selection query in ArcMap. Line 199-201. There are few other places as well. In Figure 2, the first bar shows SOV. What is it? It is not mentioned and defined anywhere in the text. Line 256: Use a reference instead of the full URL. Line 295-96: Same as above. Not a big deal but visually it would look good if maps in Figure 4 are tilted a bit left to make them proper horizontal. Figure 2: No need for extra 2 decimal places on the y-axis. Figure 3: Doesn’t really show much information. I strongly suggest replacing it with the map of census tracts. There are a number of problems with Figure 6. Classes and legend should be the same for comparison. Again there is absolutely no need to have 2 decimal places to represent miles. Add TN, AR, MS in the title of the figure and keep only commutes/total commutes and low-income commenters for a) and b). Figure 5: Add some city/county labels. Figure 7: The title of the chart is too big. Line 383-84: How did authors come up with this 3 Km, threshold? Provide a reference here. The conclusion also provides some discussion. I suggest using Discussion as a separate section and keep the conclusion small. I wonder if there was no other limitation in this study? Authors should also discuss low-income commuters, car ownership, car insurance, etc. in conjunction with distance and neighborhoods.Author Response
Please see attached file.

Reviewer 2 Report
Thank you for the opportunity to review this article. I think the topic is an important one, especially in the context of a high-poverty, highly segregated city with little relative economic growth. The results could be beneficial for transportation policymakers and economic development planners in planning for how to connect low-income groups to employment in high-impact ways.
Unfortunately, I don't think the methods used actually answer the hypothesis posed on p. 5, line 205. Testing the spatial mismatch hypothesis requires an analysis of two components: the distribution of the residential locations of a disadvantaged population and the distribution of the employment locations. The first is covered in this paper; the second is not. Commuting distance summarizes the realized travel behavior but we really want to know workers' accessibility to jobs--how many jobs they could reach that match their skill set within a given commute shed. Most analyses of the SMH use accessibility metrics as the means of analysis (see e.g. Hu (2015) as cited in the paper, Grengs (2010) "Job accessibility and the modal mismatch in Detroit", and many others). I would recommend taking this approach in a revision.
Even if the author decides to keep differences in commuting distance as the relevant question, a few items here would need to be addressed as well. First, the methods exclude commute trips within the same census tract. But in terms of land use and transportation policy goals, locating jobs near homes would be (in most cases) a desired outcome to reduce GHG from VMT. Excluding these trips introduces bias in the calculation of average commuting distance. On the opposite end, it appears that extreme trips were included in the calculation (Figure 5). It seems implausible that people are making daily commutes from Memphis to the Gulf Coast, Northwest Ark., or eastern Tenn. I suggest setting an upper bound of a realistic commuting distance--this may lead to different conclusions based on Table 4. Third, it's unclear whether the desire line (straight-line distance) was used as the commuting distance, or whether the analysis used a travel network to measure shortest-path distances. Given that, for example, freeways are disproportionately located in low-income neighborhoods and communities of color, I might suspect that it would change some distance calculations.
The data set for the analysis is stale now. CTPP 2012-2016 has been available for roughly a year now--the author may wish to update the analysis with the newer data set or compare differences in commuting patterns across the two data sets to strengthen the paper.
The paper would benefit from a significant round of editing upon revision. The methods are at once too detailed (readers don't need to know the author downloaded an Access data file or loaded data into ArcGIS, and the explanation of low-income definition seems quite complicated) and not detailed enough (why mention the G* and T autocorrelation coefficients, for example)? The flow in some places seems stilted: e.g. the introduction could benefit from subheaders when transitioning from the lit review about spatial mismatch to data availability and limitations; the lit review at places reads like a long list of study findings rather than an integrated whole; the first two sentences of section 2 are repetitive; it is a bit strange to read the data section as a long numbered list. Finally, the paper uses "we" but is a single-authored paper, so by that measure should use "I" instead.
As a last thought, I have to raise the question as to whether this paper is a right fit for the journal. None of the papers cited come from Sustainability, and though commuting distance proxies for vehicle emissions, the connection is implicit rather than explicitly stated.
Round 2
Reviewer 1 Report
The author has addressed all my comments/suggestions.
Author Response
Thank you very much for your comment.
Reviewer 2 Report
This paper is much improved from the previous version. Bringing in the employment data begins to solve my central concern that the paper did not answer the question it set out to.
Nevertheless, I'm having a hard time following the analysis. There are so many charts and tables---some with extraneous information---that it's very easy to lose the thread of the narrative. If the central question is whether there's a spatial mismatch in jobs between low-income residents and low-wage employment, I want to know these two things:
1. What is the distribution (and the average) commuting distance for low-income residents to low-wage jobs? How does that compare to the other two income/wage categories?
2. What proportion of jobs are located within that average commuting distance of (a) the typical low-wage worker and (b) low-income neighborhoods? What proportion are within specified distance bands? How does that compare with other wage types?
The second question is key to understanding the degree of mismatch. The extent to which the author can streamline the presentation of the results so that these answers rise to the top will create a more inviting paper to read and a more convincing argument. The conclusion, for example, makes it quite clear that while the average distance is shorter for low-wage workers, the proportion who commutes long distances is higher, and that's a problem. More of this succinct language and interpretation will help.
Other comments follow:
Lines 229-231 ("Thinly spread population accounts for public transit system’s inability to connect residents to jobs: Memphis was ranked 69th among the top 100 metropolitan areas (Tomer, Kneebone, Puentes, and Berube, 2011)."): Ranked 69th in what?
Figure 1: Why were the comparison metros selected?
Lines 273-274 ("(4) 2006-2010 ACS"): I would expect the ACS vintage to match the CTPP vintage. (There would be only small changes so it's not critical to update.)
Lines 302-304 ("The only one measure which is included in flow data is the total count of workers..."): Crosstabs of workers with selected demographic and travel characteristics is also included. It's worth clarifying.
Lines 434-480: I suggest removing most of this section as I mentioned in my first round of comments--it's quite repetitive and overly complicated. The reason to define low-income as earnings less than $1,250 per month is because that's what data are available.
Figure 3: Is the map a replacement for the bar chart? (It should be.) In the revised version, it's hard to tell if a figure has been replaced or not--I assume that if there are two figures directly next to each other, it is a replacement.
Table 2: The table is cut off at the margins
Figure 5 and lines 535-539: I suggest showing only the Memphis metropolitan area in the map and restricting the matching analysis to the same. The author is using an upper bound of 100 miles as the commute; hotspots in and flows to Little Rock, Nashville, etc. aren't relevant.
Tables 4, unnumbered, and 5: These can be combined into one table.
Table 3 and Figure 6 are duplicative
The paper needs another careful edit for grammar and concision.
